# A Cold Case of Equine Influenza Disentangled with Nanopore Sequencing

**DOI:** 10.3390/ani13071153

**Published:** 2023-03-24

**Authors:** Francesco Pellegrini, Alessio Buonavoglia, Ahmed H. Omar, Georgia Diakoudi, Maria S. Lucente, Amienwanlen E. Odigie, Alessio Sposato, Raffaella Augelli, Michele Camero, Nicola Decaro, Gabriella Elia, Krisztián Bányai, Vito Martella, Gianvito Lanave

**Affiliations:** 1Department of Veterinary Medicine, University of Bari, 70010 Valenzano, Italygianvito.lanave@uniba.it (G.L.); 2Office UVAC, Ministry of Health, 70122 Bari, Italy; 3Veterinary Medical Research Institute, 1143 Budapest, Hungary; 4Department of Pharmacology and Toxicology, University of Veterinary Medicine, 1400 Budapest, Hungary

**Keywords:** equine influence, sequencing techniques, animal importation

## Abstract

**Simple Summary:**

Using the Nanopore platform, we determined the whole genome sequence of an H3N8 equine influenza virus, identified from a 2005 outbreak in Apulia, Italy. The virus was closely related (>99% at the nucleotide level) in all the genome segments to viruses identified in Poland in 2005–2008 and was an inter-lineage multi-reassortant between Florida 2 and Eurasian viruses.

**Abstract:**

Massive sequencing techniques have allowed us to develop straightforward approaches for the whole genome sequencing of viruses, including influenza viruses, generating information that is useful for improving the levels and dimensions of data analysis, even for archival samples. Using the Nanopore platform, we determined the whole genome sequence of an H3N8 equine influenza virus, identified from a 2005 outbreak in Apulia, Italy, whose origin had remained epidemiologically unexplained. The virus was tightly related (>99% at the nucleotide level) in all the genome segments to viruses identified in Poland in 2005–2008 and it was seemingly introduced locally with horse trading for the meat industry. In the phylogenetic analysis based on the eight genome segments, strain ITA/2005/horse/Bari was found to cluster with sub-lineage Florida 2 in the HA and M genes, whilst in the other genes it clustered with strains of the Eurasian lineage, revealing a multi-reassortant nature.

## 1. Introduction

Equine influenza A virus (EIAV) infection causes acute disease with variable morbidity and low mortality in horses and odd-toed ungulates. EIAV is a major cause of respiratory disease in horses, able to spread rapidly among susceptible animals [1]. The influenza A virus (IAV) genome is composed of eight segments of negative-sense single-strand RNA. The virus is subtyped on the basis of the envelope-associated proteins, the hemagglutinin (HA) and the neuraminidase (NA) [2]. To date, 18 HA and 11 NA subtypes have been described in different susceptible hosts [3]. In particular, two EIAV subtypes have been identified in equines, such as H7N7 and H3N8 [1]. The last confirmed report of H7N7 EIAV in horses dates back to 1979 [4]. H3N8 EIAVs have been reported worldwide since 1963 [5]. In the phylogenetic analysis of the HA gene, H3N8 EIAVs seem to have diverged into distinct evolutionary lineages. By the end of the 1980s, two antigenically distinct lineages, American and Eurasian, have emerged. The American lineage has further evolved into Argentinian, Kentucky, and Florida sub-lineages [6,7]. The evolution of the Florida sub-lineage has led to the emergence of two clades, Florida 1 (FC1) and 2 (FC2), which are currently circulating worldwide [8,9,10,11,12,13]. FC1 viruses are endemic in North America and FC2 viruses are predominant in Europe. However, both lineages have originated outbreaks in Europe, South Africa, South America, and Asia [14]. More recently, outbreaks of FC1 and FC2 viruses have been also reported on the African continent [15,16,17].

In December 2005, EIAV infection was established as the cause of clinical respiratory disease in vaccinated horses in Apulia, Italy [18]. An acute respiratory disease occurred in a horse center managed by the Forest Service Rangers in Martina Franca. A total of 24 out of 33 animals developed acute clinical signs of febrile respiratory disease. The outbreak was seemingly related to a fair/market organized yearly for the promotion of local horse and donkey breeds, although other EIAV outbreaks were not reported in the same time span.

On the genetic characterization of the HA and NA genes, the virus from the outbreak (ITA/2005/horse/Bari) was characterized as H3N8 and was related to American viruses (A/Kentucky/5/02-like). The virus was hypothesized to have been imported into Italy due to the relocation of infected horses from a large outbreak described in 2003 in the United Kingdom [19]. In Europe, the circulation of Kentucky/5/02-like EIAV strains was first described in the United Kingdom in 2003, where outbreaks of influenza were reported regardless of the immunization status of the animal [19]. Epidemiological investigation revealed that in May 2003 Kentucky/5/02-like EIAV strains scattered from the United Kingdom as a result of the transport of horses from Newmarket to a racetrack in Rome, Italy, where an outbreak subsequently has risen [19]. Due to the genetic similarities between the HA and NA genes of strain ITA/2005/horse/Bari and H3N8 EIAV strains from the United Kingdom (A/eq/Stock-on-Trent/410956/04 and A/eq/Aboyne/410355/04), we hypothesized that the local 2005 outbreak could be somehow linked epidemiologically to the 2003 Rome outbreak, although other cases of EIAV had not been reported from May 2003 to December 2005, suggesting that the 2003 outbreak in Rome had been effectively contained. However, the epidemiological link between these two outbreaks remained unknown.

Over the last decade, novel protocols have been developed for whole genome sequencing (WGS) of viruses [20], taking advantage of next-generation sequencing (NGS) chemistries. Oxford Nanopore Technologies (ONT™) is a third-generation NSG platform, capable of producing long sequences [21]. The ONT™ platform can be used starting with a negligible investment in terms of hardware and has been quickly adopted by the academic and scientific community.

The shift from single (HA) or dual gene target (HA and NA) sequencing of IAVs to WGS using NGS has allowed us to gather relevant information on IAV evolution [22,23,24] thus also warranting re-analysis of old IAV isolates in archival collections. This provided the rationale for WGS of strain ITA/2005/horse/Bari [18].

## 2. Materials and Methods

### 2.1. Virus Cultivation

The virus was cultured in the allantoid cavities of 10-day embryonated hens’ eggs with the addition of antibiotics and antimycotics. The eggs were harvested after 3 days, and the allantoid fluid was tested for the presence of the virus by hemagglutination assay with 1% chicken erythrocytes in phosphate-buffered saline. Virus titer was also quantified using a quantitative reverse transcription polymerase chain reaction PCR (qRT-PCR) [25].

### 2.2. Genome Sequencing and Data Analysis

Using a PCR-based enrichment protocol after reverse transcription of the RNA [26], the genome of the EIAV isolate ITA/2005/horse/Bari was reconstructed. A total of 1 μg of purified PCR product from each sample was used as input of a library generated with the Ligation sequencing kit 1D SQK-LSK110 (ONT^TM^, Oxford UK). First, DNA was end-repaired (NEBNext Ultra II end repair/dA tailing kit; New England Biolabs, NEB, Ipswich, MA, USA), purified using AMPure XP beads in a ratio of 1:1 volume of beads per sample, and eluted in 30 μL of nuclease-free water. Sequencing adapters (AMx1D; ONT^TM^, Oxford UK) were ligated to the DNA (Blunt/TA ligation master mix; NEB Ipswich, MA, USA) after incubating at room temperature for 10 min. The adapter-ligated DNA library was purified with AMPure XP beads in a ratio of 1:2.5 volume of beads per sample, followed by 2 washes (Adapter bead binding buffer; ONT^TM^) and elution in 15 μL of elution buffer (ONT^TM^). The library was loaded onto a MinION flow cell (MIN106 R9.4; ONT^TM^, Oxford UK) and run via MinKNOW v.18.05.5 for 24 h. MinKNOW-generated FASTQ files containing reads reaching the quality threshold (Q score ≥7) were analyzed using Geneious Prime version 2021.2 (Biomatters Ltd., Auckland, New Zealand). The reads were mapped to a set of reference EIAV genome segments using MiniMap2 [27]. The Basic Local Alignment Search Tool Nucleotide (BLASTn) tool from NCBI (http://www.ncbi.nlm.nih.gov/blast/Blast.cgi, accessed on 28 February 2023) was employed to compare EIAV resulting consensus sequences to the GenBank database using default values to search for homologous hits.

### 2.3. Animal Influxes

The national register for animal trading (NSIS) was consulted to extract archival data of the year 2005 on the trading of horses.

### 2.4. Sequence and Phylogenetic Analyses

Sequence editing and multiple alignments were performed in Geneious Prime version 2021.2 (Biomatters Ltd., Auckland, New Zealand). The sequences were aligned with cognate EIAV strains retrieved from the GenBank database by Multiple Alignment using Fast Fourier Transform [28]. The correct substitution model parameters for the phylogenetic analyses were selected using the “Find the best protein DNA/Protein Models” tool provided by MEGA X version 10.0.5 software [29]. The evolutionary history was inferred by using the maximum-likelihood method, the Hasegawa–Kishino–Yano 2-parameter model, a discrete gamma distribution to model evolutionary rate differences among sites (6 categories), and supplying statistical support with 1000 replicates. Bayesian inference and neighbor-joining phylogenetic approaches were also explored.

The curated data sets used for phylogenetic analysis included 101 HA sequences, 98 NA sequences, and 26 complete genome sequences representative of the different lineages and clades.

### 2.5. GenBank Sequence Submission

The nucleotide sequences of 7 genome segments of strain ITA/2005/horse/Bari were deposited in GenBank under accession numbers OP919615 to OP919621. The HA gene sequence was previously obtained (EF117330) [18].

## 3. Results

The titer of the virus passage used for sequencing was 1,00X10^5^ copies (CT mean = 23.95) in qRT-PCR and 1: 256 in HA. A total of 877,687 reads were generated and after MiniMap2 assembling, the sequencing depth ranged from 35,415 to 2,421,160 (mean = 587,597) in the eight genome segments (Table 1).

In this analysis, seven genome segments of the isolate ITA/2005/horse/Bari showed the highest nucleotide (nt) identities (99.7 to 99.9% nt) to an H3N8 EIAV strain from Poland circulating during 2005 [30], whilst the HA-encoding gene shared 99.0% nt identity to a 2008 H3N8 EIAV strain from Poland [31] (Table 2).

An alignment of the HA protein was generated to inspect amino acid (aa) changes in the antigenic epitopes using reference EIAV strains (Appendix A). Overall, there were six aa changes between the isolate ITA/2005/horse/Bari and the closest relative in the HA gene, strain POL/2008/horse/Pulawy-1, but only two changes were mapped within HA epitope A at position 135 (R to G) and epitope D at position 213 (M to I). The mutation in epitope D was also observed in the reference FC2 strain GBR/2007/horse/Richmond-1, but not the change in epitope A. Three mutations were observed with respect to reference FC1 H3N8 viruses in epitope E (aa 78), B (aa 169), and D (aa 213). Moreover, seven mutations were observed with respect to the Eurasian reference strain GBR/1989/horse/Suffolk-1 in epitopes C (aa 48 and 276), B (aa 163 and 189), and D (aa 207, 213 and 244).

Based on NSIS data, the fluxes of horses (countries of origin) imported into Apulia provinces in 2005 are shown in Figure 1. Overall, 25,392 horses were imported into the Apulia region in 2005, of which 19,868 were destined to slaughter (DPA) and 5524 were non-DPA. The influxes of animals, based on their destinations (DPA and non-DPA) in the Apulia region (year 2005) and the provenience of non-DPA horses in November–December 2005 are also shown in Figure 1.

In phylogenetic analysis, based on the complete nucleotide sequences of the eight genome segments, strain ITA/2005/horse/Bari clustered within the Eurasian lineage in all the genome segments (PB2, PB1, PA, NP, NA, NS) but the HA and M (Figure 2 and Figure 3). In the phylogenetic trees based on segments 4 (HA) and 7 (M), the virus clustered with Florida 2 clade EIAV strains, suggesting an inter-lineage multi-reassortant genomic constellation.

## 4. Discussion

We applied a WGS protocol based on PCR enrichment of the IAV genome [26] and deep sequencing on the ONT^TM^ platform to generate the complete genome of an archival EIAV (Table 1). In the past, several reports have employed Sanger sequencing of single viral RNA segments such as the HA segment [32,33]. This approach did not allow us to gather relevant information on other genome segments and to decipher, for instance, the intricated origin of reassortant IAV strains.

In recent years, WGS, coupled with NGS, has become the preferred method for the high-resolution characterization of pathogens, including IAVs. Multiple protocols have been described for WGS-based typing of IAVs [34,35,36] and WGS has been applied to monitor the transmission events of IAVs in various settings [32,33,37,38,39,40] and for retrospective studies on archival samples [41,42]. This has allowed us to generate an expanding database of WGS data, useful for the analysis of novel IAV strains and also for the re-analysis of archival viruses. Protocols based on NGS allow us to generate WGS sequence data in less than 24 h, dramatically increasing the throughput and decreasing the costs. Moreover, unlike Sanger sequencing, the final sequences are a consensus obtained with a scalable depth coverage, allowing us to retain information on single nucleotide polymorphisms at the quasispecies level. Finally, sequence-independent protocols can be used to generate complete genome sequences of novel uncharacterized organisms, overcoming the limits of Sanger technology. ONT^TM^ is a third-generation NGS platform capable of producing long sequences with no theoretical read length limit [21], now declined into a number of applications. Whilst the costs for consumables can vary based on the multiplexing and chemistry among different NGS platforms, the hardware for the ONT™ platform can be acquired with a small investment, thus representing an excellent tool for low-budget research and diagnostics.

In our analysis, the isolate ITA/2005/horse/Bari appeared highly related (99.1 to 99.9% nt) in seven genome segments to an H3N8 EIAV strain from Poland circulating in 2005 [30] (Table 2) and in the HA segment to a 2008 Poland isolate.

On phylogenetic analysis with cognate gene sequences of H3N8 EIAV strains retrieved from the databases, strain ITA/2005/horse/Bari was found to cluster with FC2 viruses in the HA and M genes, whilst in the other genes it clustered with strains of the Eurasian lineage (Figure 2 and Figure 3), intermingled with Poland EIAV strains from 2005 and 2008. This unique pattern is suggestive of intra-typic (inter-lineage) reassortment between H3N8 viruses [12]. Interestingly, the closest relative of strain ITA/2005/horse/Bari, strain POL/2005/Horse/Pulawy-1, was also a reassortant with an FC2-like M gene, but, unlike the Italian strain, it retained a HA gene of the Eurasian lineage. It is rather difficult to ascertain the trajectory of the evolution of the 2005 Italian and Poland viruses, i.e., if there were sequential events of reassortment or if they represented a pool of reassortant viruses co-circulating in Europe in 2005. Unfortunately, for the closest relative of strain ITA/2005/horse/Bari in the HA gene, strain POL/2008/Horse/Pulawy-1, the complete genome sequence was not available, although the only sequenced segments (HA, M, and NS) suggest an FC2-like constellation.

Analysis of the complete genome of 82 EIAV H3N8 strains collected in the period 1963 to 2008 from Europe, Asia, and the Americas has shown that all of the trees displayed a similar phylogenetic pattern typical of other mammalian IAVs, in which viruses from consecutive seasons were linked by a main trunk lineage with short side branches stemming from it [12]. For all gene segments, the main trunk of the trees was first bifurcated when the American and Eurasian lineages appeared in the late 1980s [6]. A second bifurcation event occurred in the early 2000s, when the sub-lineages or clades FC1 and FC2 originated, both of which continue to circulate today [43] whilst the Eurasian lineage seems extinct [44]. The separation between the two clades FC1 and FC2 is evident for all viruses isolated after 2005 in all phylogenies, although inconsistencies have been observed in the phylogeny of some genome segments, i.e., PB1, PB2, PA, and HA, suggestive of intra-subtype reassortment [12]. Intra-subtype reassortment in influenza evolution has been documented for H1N1 and H3N2 human influenza viruses [45,46]. This phenomenon has been suspected to play a role in the onset of viruses with unusual epidemiological and phenotype features, such as variations in antigenic profile and/or virulence [46].

A similar pattern of evolution has been observed in a more recent study focused on the analysis of H3N8 EIAVs in Asia [47]. This study included 102 complete genomes of viruses from 1963 to 2017 and 202 HA sequences of FC1 and FC2 viruses, collected between 2003 and 2017 worldwide. In these analyses, FC1 and FC2 viruses were shown to split into various sub-clades seemingly circulating/evolving in restricted areas and occasionally spreading to other geographical areas. Noteworthy, no evidence of reassortment between FC1 and FC2 viruses was identified by the authors [47].

On visual inspection of the HA aa alignment, there were six aa changes between the isolate ITA/2005/horse/Bari and the closest relative, in this gene, strain POL/2008/horse/Pulawy-1 classified as clade FC2 but only two changes were mapped within HA epitope A at position 135 (R to G) and epitope D at position 213 (M to I). However, a number of aa changes were present in the antigenic epitopes when compared with reference strains of other lineages and sub-lineages. Three mutations were observed with respect to FC1 H3N8 viruses in epitopes E, B, and D and seven mutations were observed with respect to the Eurasian reference strain GBR/1989/horse/Suffolk-1 in epitopes C, B, and D. Changes in the epitopes of IAVs can drastically affect recognition of vaccine-induced immunity and continual monitoring of circulating viruses is required to update vaccine composition [43]. The HA epitope profile of the 2005 Italian virus was therefore highly conserved since only a single aa change was observed with respect to the reference H3N8 FC2 strain GBR/2007/horse/Richmond-1.

In summary, global surveillance activities have revealed that EIAV outbreaks are reported every year worldwide, and since the early 2000s, they have been caused by strains that belong to either FC1 or FC2 [46], whilst H3N8 viruses of Eurasian lineage are no longer detected and no longer recommended in vaccine composition [44]. However, the results of our analyses indicate that reassortant H3N8 viruses with a wide set of genes of Eurasian lineage were still circulating in Europe in 2005. Since surveillance investigations mostly collected data on the HA gene in those years, it is possible that similar reassortant viruses continued to circulate but they remained undetected. WGS analysis of archival collections could be useful to understand more in depth the extent of this phenomenon.

Since Poland is a major source of horses imported for local horse meat consumption, we interrogated the records of the national register for animal trading (NSIS) and observed that in 2005, 50.4% (12,788/25,392) of horses were imported from Poland, reaching 59.1% (1184/2003) in November and 61,4% (716/1167) in December. Of these, 19,868 (72.2%) were DPA, whilst 5524 (27.8%) were not (non-DPA). Overall, only 8.7% (478/5524) of the non-DPA horses were imported from Poland, suggesting that most animals imported from Poland were for the meat industry. Yet, finding a connection between the horses housed in the center of the Forest Service Rangers and meat horses was difficult considering the different destinations. However, in November and December 2005, 43/1184 (3.6%) and 28/716 (3.9%) non-DPA horses, respectively, were also introduced in Apulia from Poland. All these non-DPA animals were delivered to Latiano (Brindisi prefecture), only 40-km far from Martina Franca. Accordingly, we hypothesize that the 2005 EIAV outbreak in Bari was due to the re-introduction of EIAV with either DPA or non-DPA horses imported from Poland. This evidence, coupled with the genomic data, would re-enforce the hypothesis on the origin of the 2005 Italian EIAV outbreak.

## 5. Conclusions

Coupling molecular and epidemiological data, a 2005 cold case, i.e., the origin of an outbreak of EIAV, could be finally disentangled and the 2005 H3N8 EIAV strain was found to be a multi-reassortant between Eurasian lineage and FC2 viruses.

WGS is now being applied more routinely to EIAV analysis [26], thus helping understand the evolutionary routes of H3N8 EIAV strains on a global scale [12]. In addition, since H3N8 EIAVs have also been identified in dogs [48], pigs, cats [49,50], and in camels [51], this information is useful to understand promptly and thoroughly inter-species transmission events. Although the evolution of H3N8 EIAV is almost entirely driven by mechanisms of genetic drifts [52], WGS has revealed that reassortment may also occur between H3N8 EIAV strains and with IAVs of non-equine origin [12,53]. Genome segments related to influenza A H1N8, H5N1, H7N1, and H9N2 strains have been identified among samples collected from horses, accounting for less than 0.5% of all EIAV cases [54]. Moreover, a novel H9N2 influenza A virus was detected in 2011, in horses in Guangxi Province, China [55]. Generating a large database of EIAV genome sequences will surely improve the resolution of epidemiological investigations. Understanding the molecular evolution and phylodynamic of EIAV requires a large-scale and comprehensive genome data set. This should also include the analysis of archival EIAV strains.

## Figures and Tables

**Figure 1 animals-13-01153-f001:**
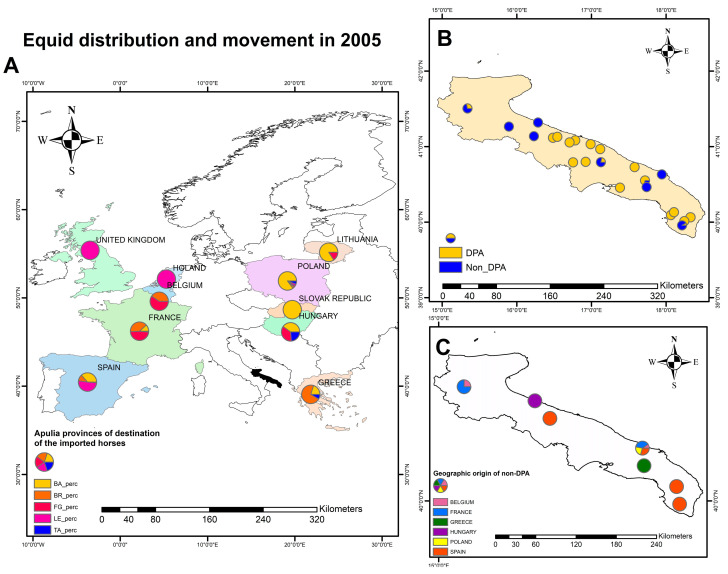
Fluxes of horses (countries of origin) imported in Apulia provinces (BA, Bari; BR, Brindisi; FG, Foggia; LE, Lecce; TA, Taranto) in 2005 (**A**); influxes of animals, based on their destinations (DPA, animals destined to slaughter; non-DPA, animals with other destination) in Apulia region (**B**); geographic provenience of non-DPA horse in November-December 2005 (**C**). Pie charts are not proportionate to the number of animals.

**Figure 2 animals-13-01153-f002:**
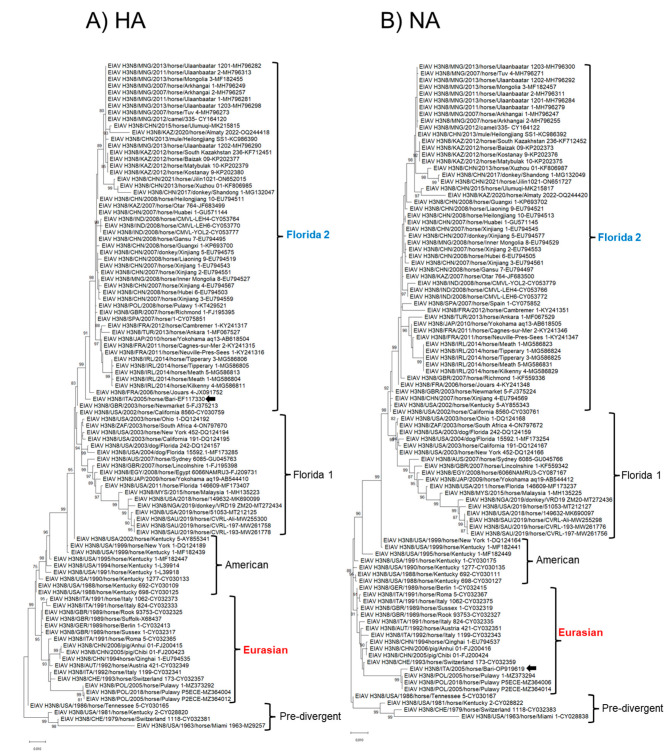
Phylogenetic analysis of haemagglutinin and neuraminidase genes of EIAV subtype H3N8 strain ITA/2005/horse/Bari compared with cognate sequences retrieved from GenBank database. Phylogeny of the hemagglutinin (HA) gene (**A**). Phylogeny of the neuraminidase (NA) (**B**). Phylogenetic trees were performed using the maximum likelihood method and Hasegawa–Kishino–Yano 2-parameter model with a gamma distribution. A total of 1000 bootstrap replicates were used to estimate the robustness of the individual nodes on the phylogenetic tree. Bootstrap values greater than 75% were indicated. Black arrow indicates the strain described in this study. Numbers of nucleotide substitutions are indicated by the scale bar.

**Figure 3 animals-13-01153-f003:**
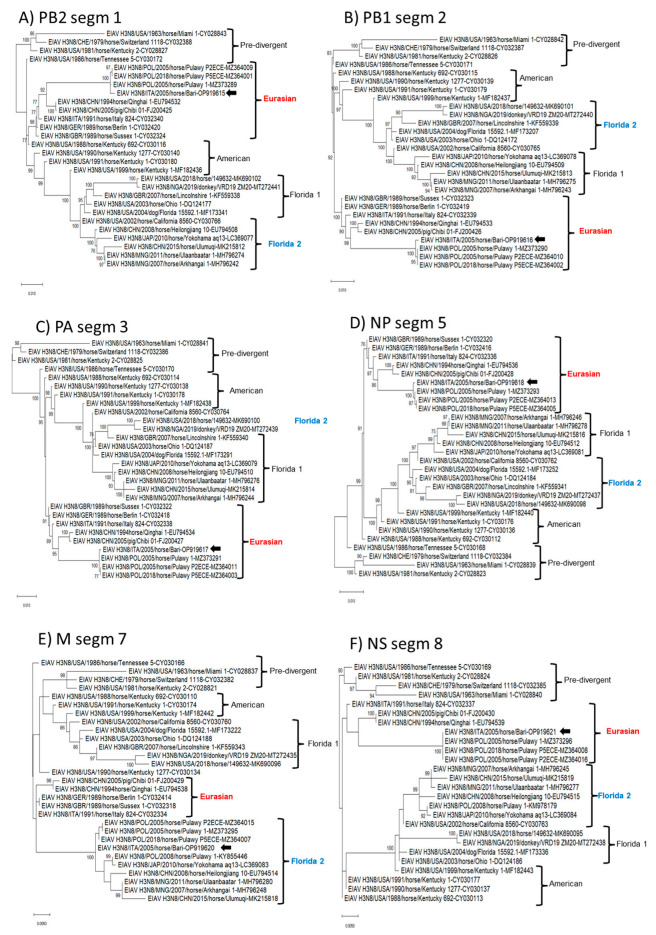
Phylogenetic tree of influenza virus polymerase complex (Polymerase basic protein 2, PB2, Polymerase basic protein 1, PB1, Polymerase acidic, PA), nucleoprotein (NP), matrix protein (M), and non-structural gene (NS) nucleotide sequences of the EIAV subtype H3N8 strain ITA/2005/horse/Bari were compared with cognate sequences retrieved from GenBank database. Phylogeny of PB2 gene (**A**). Phylogeny of PB1 gene (**B**). Phylogeny of PA gene (**C**). Phylogeny of NP gene (**D**). Phylogeny of M gene (**E**). Phylogeny of NS gene (**F**). Phylogenetic trees were performed using the maximum likelihood method and Hasegawa–Kishino–Yano 2-parameter model with a gamma distribution. A total of 1000 bootstrap replicates were used to estimate the robustness of the individual nodes on the phylogenetic tree. Bootstrap values greater than 75% were indicated. Black arrow indicates the strain described in this study. Numbers of nucleotide substitutions are indicated by the scale bar.

**Table 1 animals-13-01153-t001:** Number of reads obtained and depth of coverage per genome segment for strain A/equine/Bari/2005 by Minimap2.

Gene	Segment	Length (nt)	Reads (Number)	Coverage Depth	Accession
PB2	1	2341	16,195	35415.1	OP919615
PB1	2	2341	19,245	42029.6	OP919616
PA	3	2233	28,177	63306.4	OP919617
HA	4	1778	14,653	42886.6	EF117330
NP	5	1565	26,191	84524.7	OP919618
NA	6	1413	41,878	153256.3	OP919619
M	7	1027	360,563	1858200.5	OP919620
NS	8	890	409,258	2421160.6	OP919621

Polymerase basic protein 2 (PB2), Polymerase basic protein 1 (PB1), Polymerase acidic (PA), Hemagglutinin (HA), Nucleoprotein (NP), Neuraminidase (NA), Matrix (M), Non-structural protein (NS).

**Table 2 animals-13-01153-t002:** Equine influenza viruses with the highest nucleotide identity to ITA/2005/horse/Bari, determined by Basic Local Alignment Search Tool Nucleotide (BLASTn) (consulted on 28 February 2023).

Gene	Gene Accession No.	Viruses with Highest % of Nucleotide Identity	Origin	Percent Identity
PB2	MZ373289	A/equine/Pulawy/1/2005(H3N8)	Poland	99.7
PB1	MZ364002	A/equine/Pulawy/1/2005(H3N8)	Poland	99.8
PA	MZ373291	A/equine/Pulawy/1/2005(H3N8)	Poland	99.9
HA	KT429521	A/equine/Pulawy/1/2008(H3N8)	Poland	99.0
NP	MZ373293	A/equine/Pulawy/1/2005(H3N8)	Poland	99.9
NA	MZ373294	A/equine/Pulawy/1/2005(H3N8)	Poland	99.8
M	MZ373295	A/equine/Pulawy/1/2005(H3N8)	Poland	99.8
NS	MZ373296	A/equine/Pulawy/1/2005(H3N8)	Poland	99.9

Polymerase basic protein 2 (PB2), Polymerase basic protein 1 (PB1), Polymerase acidic (PA), Hemagglutinin (HA), Nucleoprotein (NP), Neuraminidase (NA), Matrix (M), Non-structural protein (NS).

## Data Availability

The data presented in this study are available on request from the corresponding author.

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
