# Peer review of "A Cold Case of Equine Influenza Disentangled with Nanopore Sequencing"

_animals, 2023, doi:10.3390/ani13071153_

Round 1
Reviewer 1 Report
The authors describe the isolation of a H3N8 equine influenza virus identified from a 2005 outbreak in Italy closely related to one circulating in Poland.
The report is interesting but lacks of several analyses and a better presentation of the data.
1) There are many sequences for H3N8 equine influenza virus available in literature, authors should make a tree since they have the full genome
2) The authors present few bits of genomic data without going into detail of what the mutations are in the new genome compared to the close Poland one.
3) Need to show more blast hits to know that the analyses only show Poland strain as closest.
4) Can authors show an alignment of similarity between the two strains?
5) Table 2 is hard to interpret and it’s not in the result section
6) Methods : why virus was cultured in eggs? That is odd. It’s a very old method known (for other viruses at least) to produce mutations and attenuate the virus. Have authors tried to sequence the virus from the sample without in vitro passage? Wouldn’t be better an equine lung cell line instead if needs for in vitro passage?
Author Response
Reviewer 1
The authors describe the isolation of a H3N8 equine influenza virus identified from a 2005 outbreak in Italy closely related to one circulating in Poland.
The report is interesting but lacks of several analyses and a better presentation of the data.
R1.1 There are many sequences for H3N8 equine influenza virus available in literature, authors should make a tree since they have the full genome
Reply to R1.1 We agree with the referee on this observation. Accordingly, we performed phylogenetic analyses on the 8 different gene segments of the equine influenza virus H3N8 genome using cognate sequences of H3N8 equine viruses available in the databases. We provided detailed information in the manuscript and we added two novel figures (Figure 2 and 3). Based on this analysis, the Italian 2005 virus seems an inter-lineage multi reassortant. This information was added and discussed.
R1.2 The authors present few bits of genomic data without going into detail of what the mutations are in the new genome compared to the close Poland one.
Reply to R1.2. As suggested by Referee 2, we focused on the mutations in the HA gene and we added a supplemental figure with HA alignment and commented this shortly in the discussion.
R1.3 Need to show more blast hits to know that the analyses only show Poland strain as closest.
Reply to R1.3 As suggested by the referee also in point R1.1, we implemented the manuscript with phylogenetic analyses based on all the genome segments (Figure 2 and 3). The virus ITA/2005/Horse/Bari was actually a inter-lineage multi reassortant with segments 7 (M) and 4 (HA) of Florida 2 sublineage and the other genome segments of Eurasian lineage, intermingled in all the figures with Poland strains. We hope that this representation of data (as suggested by Referee 2) could fulfill the request of Referee 1. Also, please note that for editorial/formatting constrains it is difficult to edit table 1, adding to much information.
R1.4 Can authors show an alignment of similarity between the two strains?
Reply to R1.4 As replied in R1.3 we have provided this information in the form of phylogenetic analyses since this presentation of data is more informative for the readers (Figure 2 and 3). The virus ITA/2005/Horse/Bari was actually a inter-lineage multi reassortant with segments 7 (M) and 4 (HA) of Florida 2 sublineage and the other genome segments of Eurasian lineage.
An alignment was generated for the HA epitopes (supplemental figure) (see also reply to point R1.2).
R1.5 Table 2 is hard to interpret and it’s not in the result section
Reply to R1.5 We moved Table 2 to the result section and in the light of figure 2 and 3 we think that the table now is more informative.
R1.6 Methods: why virus was cultured in eggs? That is odd. It’s a very old method known (for other viruses at least) to produce mutations and attenuate the virus. Have authors tried to sequence the virus from the sample without in vitro passage? Wouldn’t be better an equine lung cell line instead if needs for in vitro passage?
Reply to R1.6 Unfortunately, the original sample was not available and we could only analyze the isolate obtained on eggs. We used the embryonated eggs since specific cells for EIAV were not available in our lab. Yet, we agree with the referee that other substrates are available for virus cultivation and that mutation can occur during adaptation. Please consider that our study was not intended to study particular mutations since we do not have a virus pair (original sample and virus isolated in vitro) but rather to understand the origin of the virus outbreak.
Reviewer 2 Report
The manuscript entitled "A cold case of equine influenza disentangled with Nanopore sequencing" by Pellegrini et al. describes the whole genome sequencing of an equine H3N8 influenza virus that was isolated in 2005 from horses in Italy.
Fundamentally, the manuscript details the amplification of an influenza virus isolate in embryonated chicken eggs, the sequencing of all eight virus gene segments, and the comparison of these sequences against genome data in GenBank. This approach has been employed for many years in influenza research and has been instrumental to our understanding of influenza evolution and epidemiology.
In contrast to the Sanger sequencing method employed traditionally, the authors used a whole genome sequencing technique employing the nanopore sequencing platform. By comparing the nanopore sequencing data obtained to genome data available in GenBank, the authors were able to conclude that the eight gene sequences of A/eq/Bari/2005 were very closely related to a virus that had circulated in horses in Poland during the same time period. Therefore, the novelty of the manuscript lies within the nanopore sequencing technique employed and not necessarily in the finding that the virus originated in horses imported from Poland (this conclusion could also have been drawn with traditional sequencing). In light of this, the authors might want to consider adding some discussion of the nanopore methodology, such as pros and cons versus Sanger (e.g., ease of use, turnaround time, cost, quality of sequence etc.).
There are some additional (minor) issues with the manuscript that warrant attention:
1. Some minor English language edits are required.
2. Introduction, line 43: While H3N8 viruses occasionally have been isolated from human volunteers, the significance of this should not be overstated. Please consider deleting the sentence.
3. Results, line 110: Please define acronyms upon first usage (e.g., DPA).
4. Discussion, line 157: The authors state that the influenza outbreak in Italy was "not related epidemiologically to the 2003 horse racing outbreak in Rome." Please clarify or delete the statement (i.e., if the Polish virus be traced to the virus in the U.K. then the statement is incorrect).
5. Figure 1: The numbers next to the circles in figure 1A are difficult to read and it is unclear what they mean. Please delete or add a description to the figure legend.
Author Response
Reviewer 2
The manuscript entitled "A cold case of equine influenza disentangled with Nanopore sequencing" by Pellegrini et al. describes the whole genome sequencing of an equine H3N8 influenza virus that was isolated in 2005 from horses in Italy.
Fundamentally, the manuscript details the amplification of an influenza virus isolate in embryonated chicken eggs, the sequencing of all eight virus gene segments, and the comparison of these sequences against genome data in GenBank. This approach has been employed for many years in influenza research and has been instrumental to our understanding of influenza evolution and epidemiology.
R2.1. In contrast to the Sanger sequencing method employed traditionally, the authors used a whole genome sequencing technique employing the nanopore sequencing platform. By comparing the nanopore sequencing data obtained to genome data available in GenBank, the authors were able to conclude that the eight gene sequences of A/eq/Bari/2005 were very closely related to a virus that had circulated in horses in Poland during the same time period. Therefore, the novelty of the manuscript lies within the nanopore sequencing technique employed and not necessarily in the finding that the virus originated in horses imported from Poland (this conclusion could also have been drawn with traditional sequencing). In light of this, the authors might want to consider adding some discussion of the nanopore methodology, such as pros and cons versus Sanger (e.g., ease of use, turnaround time, cost, quality of sequence etc.).
Reply to R2.1. This topic was briefly already mentioned in the discussion. We included some additional elements, mentioning the advantages of massive sequencing (see page 8 lines 227-233). However, we would like to point out that we also generated the complete genome sequence of a EIAV strain, and understood that the virus ITA/2005/Horse/Bari was actually a inter-lineage multi reassortant with segments 7 (M) and 4 (HA) of Florida 2 sublineage and the other genome segments of Eurasian lineage.
There are some additional (minor) issues with the manuscript that warrant attention:
R2.2. Some minor English language edits are required.
Reply to R2.2. The English has been revised throughout the manuscript.
R2.3. Introduction, line 43: While H3N8 viruses occasionally have been isolated from human volunteers, the significance of this should not be overstated. Please consider deleting the sentence.
Reply to R2.3. As suggested by the referee, this sentence was deleted.
R2.4. Results, line 110: Please define acronyms upon first usage (e.g., DPA).
Reply to R2.4. This was corrected (page 4, line 160).
R2.5. Discussion, line 157: The authors state that the influenza outbreak in Italy was "not related epidemiologically to the 2003 horse racing outbreak in Rome." Please clarify or delete the statement (i.e., if the Polish virus be traced to the virus in the U.K., then the statement is incorrect).
Reply to R2.5. As suggested by referee 2, this part of the discussion was deleted.
R2.6. Figure 1: The numbers next to the circles in figure 1A are difficult to read and it is unclear what they mean. Please delete or add a description to the figure legend.
Reply to R2.6. The numbers next to the circles in figure 1A were deleted.
Round 2
Reviewer 1 Report
authors replied to comments. no further questions.